# Revealing the Regulatory Mechanism of lncRNA-LMEP on Melanin Deposition Based on High-Throughput Sequencing in Xichuan Chicken Skin

**DOI:** 10.3390/genes13112143

**Published:** 2022-11-17

**Authors:** Pengwei Zhang, Yanfang Cao, Yawei Fu, Huiyuan Zhu, Shuohui Xu, Yanhua Zhang, Wenting Li, Guirong Sun, Ruirui Jiang, Ruili Han, Hong Li, Guoxi Li, Yadong Tian, Xiaojun Liu, Xiangtao Kang, Donghua Li

**Affiliations:** 1College of Animal Science and Technology, Henan Agricultural University, Zhengzhou 450046, China; 2Henan Innovative Engineering Research Center of Poultry Germplasm Resource, Zhengzhou 450046, China

**Keywords:** skin pigmentation, RNA-seq, melanin, chicken, lncRNA-LMEP

## Abstract

The therapeutic, medicinal, and nourishing properties of black-bone chickens are highly regarded by consumers in China. However, some birds may have yellow skin (YS) or light skin rather than black skin (BS), which causes economic losses every year. Long noncoding RNAs (lncRNAs) are widely present in living organisms, and they perform various biological functions. Many genes associated with BS pigmentation have been discovered, but the lncRNAs involved and their detailed mechanisms have remained untested. We detected 56 differentially expressed lncRNAs from the RNA-seq of dorsal skin (BS versus YS) and found that *TCONS_00054154* plays a vital role in melanogenesis by the combined analysis of lncRNAs and mRNAs. We found that the full length of the *TCONS_00054154* sequence was 3093 bp by RACE PCR, and we named it *LMEP*. Moreover, a subcellular localization analysis identified that *LMEP* is mainly present in the cytoplasm. After the overexpression and the interference with *LMEP*, the tyrosinase content significantly increased and decreased, respectively (*p* < 0.05). In summary, we identified the important lncRNAs of chicken skin pigmentation and initially determined the effect of *LMEP* on melanin deposition.

## 1. Introduction

As an essential genetic and economic trait, chicken skin color has long been an important research topic that has been studied by scientists. The therapeutic, medicinal, and nourishing health benefits of black-bone chickens have been highly regarded in oriental countries for several thousand years [1]. When consumers buy black-bone chickens, they mainly observe the color of the chicken skin with the naked eye, and they preferentially purchase black-bone chickens with black skin [2]. The Xichuan black-bone chicken, formerly known as the medicinal chicken, is produced in Xichuan County, Henan Province, and it is even considered to be one of China’s rare local breed resources. This chicken possesses a black beak, skin, bones, legs, and meat [3]. However, during the breeding process, some produced chickens may have yellow skin, which slows the progress of selective breeding and causes economic losses for poultry farmers [2].

Skin color is one of the most critical phenotypes of poultry in nature. It is closely related to environmental adaptability, nutritional conditions, and species identification. Black-bone chickens are known because of their black skin, exhibiting more melanin hyperpigmentation than other standard chickens [4,5]. Skin pigmentation undergoes a complex series of biological processes, mainly melanogenesis and keratinocyte migration [6,7]. The research on tyrosinase, which is also known as monophenol monooxygenase, shows that mutations in the tyrosinase gene can lead to abnormal skin pigmentation, and this can result in skin diseases [8]. As a key enzyme in melanin synthesis, tyrosinase plays a crucial role in the complete process of its production, and the activity and quantity of tyrosinase directly affects the melanin production [9]. The complex mechanism of pigment deposition is influenced by nutritional, physiological, genetic, environmental, and pharmaceutical factors, with genetics playing a dominant role in this [10,11].

With the prevalence and improvement of sequencing technology, an increasing number of lncRNAs have been discovered to be biologically meaningful in various organisms. Long noncoding RNAs (lncRNAs) are noncoding RNAs that are more than 200 nucleotides in length and may be located in different subcellular regions, such as the chromatin, nucleus, and cytoplasm [5,12,13]. lncRNAs act as signaling, decoy, guide, or scaffold molecules to regulate gene expression, and they are involved in development, cell differentiation, and various metabolic processes at the epigenetic and transcriptional levels [14,15]. However, to our knowledge, the effect of lncRNAs on chicken skin pigmentation is not well known in transcriptomic studies that are related to chicken skin pigmentation. Black-skinned and yellow-skinned Xichuan black-bone chickens occur within the same breed, but there are apparent differences in the skin color traits. There has been no report in the research that is related to this phenomenon, but these chickens are an ideal model to study the lncRNAs that regulate BS pigmentation.

In the past few years, lncRNAs have been found to regulate various biological processes by acting in *cis* or *trans* [16,17] fashion, and they can also use various mechanisms to affect gene expression. Many studies suggest that lncRNAs can regulate melanin deposition. Bhat et al. [18] identified that 57 differentially expressed lncRNAs are mainly involved in the pathways of melanin biosynthesis and transport functions by the transcriptome sequencing of cape goat skin samples of different colors. Similarly, three melanin-related lncRNAs, Ccr_lnc142711, Ccr_lnc17214525, and Ccr_lnc14830101, were identified by sequencing in koi carp with varying colors of skin (black, white, and red) [19]. Dan et al. [20] induced mouse melanoma B16 cells with all-trans retinoic acid and 4-phenylbutazone and found a significant decrease in lncRNA-GM31932 by conducting a whole transcriptome analysis. Further study revealed that lncRNA-GM31932 could decrease the *Prc1* and *Nuf2* expression, induce the cell cycle, and increase the melanin production. Another study found a negative correlation between lncRNA UCA1 and the melanin content. *UCA1* regulates the CREB/MITF axis through the CREB/PKA and ERK/JNK signaling pathways, and it inhibits melanogenesis [21]. All of the above experiments demonstrate that lncRNAs play an essential role in regulating melanin deposition. However, the functional and molecular mechanisms of lncRNAs have yet to be documented in the research regarding pigmentation in the chicken skin transcriptome.

In this work, we first measured the dorsal skin color of each chicken using an NR10QC portable colorimeter, and we used H and E staining for the histological examination. Then, we examined the lncRNA expression profiles in BS and YS to investigate lncRNAs that are involved in skin pigmentation. We screened the core lncRNAs by analyzing the differentially expressed lncRNA target genes and mRNAs, and we performed bioinformatics analysis and biological function studies on the core lncRNA-LMEP (Figure 1). In summary, this work predicts the function of the lncRNAs, and these predictions can provide insight for further research on the regulation of chicken skin pigmentation by lncRNAs. The study of the biological process of *LMEP* also lays the foundation for studying the role of the lncRNAs in melanogenesis.

## 2. Materials and Methods

### 2.1. Ethics Statement

The chicken experiments were conducted in strict accordance with the requirements of the Institutional Animal Care and Use Committee (IACUC) of Henan Agricultural University, China (11-0099). The chickens were humanely sacrificed to reduce their suffering.

### 2.2. Sample Preparation

We acquired the Xichuan black-bone chickens from the Henan Agricultural University chicken farm (Zhengzhou, China). Six healthy female chickens with different skin color phenotypes were selected for the experiment, and they were divided into two groups: the YS pigmentation phenotype group (n = 3) and the BS pigmentation phenotype group (n = 3). A determination of the dorsal skin color of each chicken was performed using an NR10QC portable colorimeter (3nh, China). The dorsal skin samples (tissue with an area of 1.5 × 1.5 cm between the two dorsal wings) were harvested from the chickens on day one and rapidly fixed in 4% formalin, paraffin-embedded, and subjected to ordinary H and E staining for the histological examination. Approximately 0.5 cm^2^ of dorsal skin was collected, rapidly placed in liquid nitrogen, and subsequently stored at −80 °C, and this was used for the RNA-seq and the quantitative real-time PCR (qRT-PCR) analysis.

### 2.3. Total RNA Isolation and Illumina Sequence Analysis

The total RNA was extracted from the chicken dorsal skin tissue using TRIzol reagent (Invitrogen, USA). The purity and quantity of RNA were measured using a NanoPhotometer spectrophotometer (Implen, Munich, Germany). The integrity was assessed using an RNA Nano 6000 Assay Kit with an Agilent Bioanalyzer 2100 system (Agilent Technologies, Santa Clara, CA, USA). The samples were then sequenced by Biomarker Technologies Corporation (Beijing, China) on an Illumina HiSeq 4500 with paired-end (PE) 150 bp sequencing. The reads were cleaned and analyzed according to the manufacturer’s instructions (Illumina, San Diego, CA, USA). The raw reads were filtered with in-house Perl scripts to obtain mappable reference genome sequences using the TopHat2 tools [22]. The mapped reads were assembled using Scripture (beta2) [23] and Cufflinks (v2.1.1) [24]. The reads per kilobase per million mapped reads (RPKM) values were used to quantify the normalized reads. The lncRNAs were annotated with references using the NONCODE database (http://www.noncode.org/, accessed on 5 March 2018). The coding potential of RNA transcripts that were longer than 200 nt with open reading frames (ORFs) that were less than 300 nt in length and FPKM values that were greater than 0.01 was predicted by using the Coding-Non-Coding Index (CNCI), the Coding Potential Assessment Tool (CPAT), the Coding Potential Calculator (CPC), and the Pfam protein family database [25,26,27,28]. The differentially expressed lncRNAs were identified by using Cuffdiff (v2.1.1) [24]. A false discovery rate (FDR)-corrected *p* value < 0.05 and a |log2 (fold change)| ≥ 1 were considered to indicate a differential expression between the two groups of chickens (BS and YS). The sequencing data were submitted to the Sequence Read Archive (SRA) (accession number: PRJNA418694) of the National Center for Biotechnology Information (NCBI).

### 2.4. Target Gene (cis and trans) Prediction and Enrichment Analysis

We performed the target gene prediction on the differentially expressed lncRNAs. We identified *cis* target genes by searching for genes encoding 100 kb upstream and downstream of the lncRNAs. The *trans* lncRNA target genes were identified based on their expression levels. The GO enrichment analysis of the target genes was implemented with the GOseq R package based on Wallenius’ noncentral hypergeometric distribution [29], and the KEGG enrichment analysis of the differentially expressed lncRNA target genes was performed using the KOBAS software [30]. All of the heatmaps for gene clustering in the present study were constructed using the R program (http://www.R-project.org/, accessed on 12 March 2018).

### 2.5. Rapid Amplification of cDNA Ends (RACEs)

Based on RNA-seq, we obtained partial *LMEP* sequences. We obtained the full-length *LMEP* sequence by conducting RACE PCR. The total RNA from the dorsal skin tissue was used as a template for the nested PCR using the SMARTer RACE cDNA Amplification Kit (TaKaRa, Dalian, China) following the manufacturer’s instructions. The RACE PCR products were cloned into the pMD™18-T Vector (TaKaRa, Dalian, China) and sequenced. The RACE PCR primer information is shown in Appendix A.

### 2.6. LMEP Subcellular Localization of Melanocytes

In this study, the PARIS kit (Life Technologies, Carlsbad, CA, USA) was used to isolate the nucleus and cytoplasm of the melanocytes. The cell suspension was collected and centrifuged at 1000× *g* for 10 min. After washing, a precooled Cell Fractionation Buffer was added, and then, it was incubated on ice for 10 min, and centrifuged at 1000× *g* at 4 °C for 5 min. The nucleus (sediment fraction) and cytoplasm (supernatant) were obtained. The nucleus was lysed with Cell Disruption Buffer, and the nucleus and cytoplasm were washed according to the kit requirements. We detected the expression of *LMEP* in the nucleus and cytoplasm of the melanocytes by the qRT-PCR, and *Sno-u6* and *GAPDH* were used as internal reference genes for the nuclear and cytoplasmic RNA quantification, respectively (*Sno-u6* and *GAPDH* are stably expressed in melanocytes).

A Fluorescence In Situ Hybridization kit (Ribobio, Guangzhou, China) was used to conduct the *LMEP* fluorescence during the in situ hybridization. The cell slide was placed at the bottom of the 6-well plate. When the cell confluence reached 60–70%, the cells were fixed with 4% paraformaldehyde at room temperature for 30 min. After washing, the precooled 0.3% Triton X-100 was added, and the cells were rested at 4 °C for 30 min. The prehybridization solution was added, and the cells were sealed at 37 °C for 30 min. The prehybridization solution was discarded, and a probe hybridization solution containing *LMEP* FISH Probe Mix was added. The cells were washed 3 times at 42 °C for 5 min each to reduce the background signal, then they were incubated with DAPI staining solution for 10 min at room temperature, and they were washed with PBS (phosphate-buffered saline) 3 times for 5 min each, which was followed by the fluorescence detection. All of the procedures were performed under light-proof conditions.

### 2.7. Bioinformatics Analysis

The locations of *LMEP* on the genome and the distribution of exons were predicted using UCSC (http://genome.ucsc.edu, accessed on 10 June 2018). We predicted the coding ability of the chicken *LMEP* sequences using CPC (http://cpc2.cbi.pku.edu.cn, accessed on 10 June 2018) and CPAT (http://lilab.research.bcm.edu/cpat/index.php, accessed on 10 June 2018). The subcellular molecular localization of *LMEP* was predicted using lncLocator (http://www.csbio.sjtu.edu.cn/bioinf/lncLocator/, accessed on 10 June 2018). We predicted the lncRNA secondary structure using AnnoLnc (http://annolnc.cbi.pku.edu.cn, accessed on 10 June 2018).

### 2.8. Plasmid Construction and RNA Interference

According to the results of the RACE PCR experiment, the primers with the NheI and AgeI restriction enzyme sites were designed at each end of *LMEP* to enable the amplification of the entire length of *LMEP* and its ligation into the pcDNA3.1-EGFP vector. The primer sequences are shown in Appendix A. The siRNA for *LMEP* was designed and synthesized using GenePharma (Shanghai, China). The primer sequence information is shown in Table 1.

### 2.9. Isolation and Purification of Chicken Melanocytes

The chicken melanocytes were isolated from 20 embryonic Xichuan black-bone chickens. Firstly, the eggs were wiped with 75% alcohol, the embryos were removed from the side of the air chamber, and the peritoneum was obtained by excision. The peritoneum was placed in prewarmed PBS with double antibodies at 37 °C, and then, they were washed to remove grease and other impurities (this was repeated three times). The cleaned peritoneum was cut into pieces using ophthalmic scissors and washed three times with PBS, then, they were placed into a 50 mL centrifuge tube with Medium 254 (Gibco, New York, NY, USA) before the addition of an appropriate amount of Proteinase II enzyme and 0.25% trypsin digestion solution (1:1); they were digested at 37 °C for 60 min (the suspended tissue floats and does not sink easily, the lump becomes flocculent when digestion is completed), and the addition of a complete medium terminated the digestion. After repeated blowing, the samples were passed through a 200 + 400 mesh stainless steel sieve, and the filtrate was collected in a centrifuge tube. The samples were centrifuged at 1000 r/min for 10 min, and the supernatant was discarded. Medium 254 was added, which was mixed with gentle blowing, and they were centrifuged at 1000 r/min for 10 min, and the supernatant was removed (repeated 3 times). Fresh medium containing HMGS (Gibco, USA) (HMGS:Medium 254 = 1:100) was added, mixed well, and inoculated into cell bottles. The plates were incubated in 5% CO_2_ at 37 °C.

When the melanocytes grew to 70–80% confluence, the medium was discarded, the cells were washed with PBS 3 times, an appropriate amount of digestive solution (0.25% trypsinase-0.02% EDTA) was added, and the cells were digested at 37 °C for 7–10 min (when the cells were digested to approximately a half-rounded shape, complete medium was added to terminate digestion). After the centrifugation at 1000 r/min for 10 min, the supernatant was discarded, and fresh medium containing HMGS was added and cultured in an incubator with 5% CO_2_ and 37 °C. The solution was changed every 2 days.

### 2.10. Cell Culture and Transfection

The primary melanocytes were identified by Dopa staining, immunofluorescence, and transmission electron microscopy (Appendix A). The melanocytes were grown at 37 °C and 5% CO_2_, and they were cultured in Medium 254 (Gibco, USA) containing 1% HMGS (Gibco, USA). All of the transient transfections were performed using Lipofectamine 2000 reagent (Invitrogen, Carlsbad, CA, USA), and the cells were collected after 48 h of incubation.

### 2.11. Detection of Tyrosinase Content

The tyrosinase content was measured using an ELISA kit from Nanjing Jiancheng Bioengineering Institute (Nanjing, China). The experiment included a blank well, a standard well, and a sample well, and each group included three replicates. Only the chromogenic agent and stop solution were added to the blank well. Fifty microliters of standard and sample were added to standard and sample wells, respectively, which was followed by 50 µL of HRP, and it was incubated at 37 °C for 60 min. After discarding the solution, 200 µL of washing solution was added, and then, it was washed for 30 s and patted dry, which was repeated 5 times. After adding the chromodeveloper, the treatment was kept in the dark at 37 °C for 10 min, and 50 µL of the stop solution was added to terminate the treatment. The OD value was detected at a wavelength of 450 nm.

### 2.12. qRT-PCR

The quantitative primers were designed by Primer 6.0 software (*β-actin* acts as an internal reference gene and it is stably expressed in melanocytes), and the sequences are listed in Appendix A. The qRT-PCR was performed with 10 μL reaction mixtures containing 1.0 µL of cDNA, 5 μL of SYBR Premix Ex Taq Ⅱ (Tli RNase H Plus) (TaKaRa, Dalian, China), 0.5 µL of each the forward and universal downstream primers primer (10 μM), and 3 µL of H_2_O. The reaction strategy mainly consisted of pre-denaturation (95 °C for 5 min), denaturation (95 °C for 15 s), annealing (60 °C for 45 s), and extension (72 °C for 40 s), with a total of 35 cycles.

### 2.13. Statistical Analysis

The results were expressed as mean ± SD (standard deviation) and analyzed by one-way ANOVA using SPSS (version 26.0, USA), and the differences were considered to be significant at *p* value < 0.05.

## 3. Results

### 3.1. RNA-Seq of Dorsal Skin

There were apparent differences in the skin pigmentation phenotypes among the six Xichuan black-bone chickens. The color values are recorded in Appendix A. We found more melanin granules by H and E staining in the BS than we did in the YS (Figure 2). To better explore the mechanism of the lncRNAs during the melanogenesis in the chicken skin, we set up six libraries for RNA-seq. After removing the low-quality and adaptor sequences, we obtained 22,046,187–24,387,777 clean reads with Q30 values ranging from 86.07–86.66%, and the percentages of mapped reads ranged from 69.14–70.57% (Appendix A). These results indicated that our six libraries could be used to compare the skin transcriptomes from the chickens with the YS and BS pigmentation phenotypes.

### 3.2. Genomic Features of lncRNAs

A total of 1089 lncRNAs from the chicken skin tissue were identified and screened using four analytical tools: the CNCI, CPAT, CPC, and Pfam (Figure 3a). Many novel lncRNAs were present in the YS and BS ones. The novel lncRNAs comprised 82.989% long intergenic ncRNAs (lincRNAs), 3.245% sense lncRNAs, and 13.766% antisense lncRNAs. The known lncRNAs comprised 85.333% intergenic lincRNAs and 14.667% antisense lncRNAs (Figure 3b). The mRNA and lncRNA expression profiles were compared based on the density profiles of the fragments per kb per million reads (FPKM) for all of the transcripts, and the expression levels of lncRNAs were lower than those of the mRNAs (Figure 3c). The lncRNAs were alternatively spliced with 1.13 isoforms per lncRNA locus, and the mRNAs were also alternatively spliced with 1.42 isoforms per mRNA locus (Figure 3d).

### 3.3. Cluster and Differential Expression Analysis of lncRNAs

To more thoroughly evaluate the differentially expressed lncRNAs, verify the repeatability of the data, and identify similar expression patterns, a hierarchical clustering analysis of the differentially expressed lncRNAs identified from the six libraries was performed. The aggregation of the different samples or tissues is a crucial method in a high-throughput data analysis. All of the differentially expressed lncRNAs that were identified in the dorsal skin tissues of BS and YS groups (each with three biological replicates) of the Xichuan black-bone chickens were analyzed by a cluster analysis based on the obtained FPKM values (Figure 3e). The six libraries clustered well into the BS and YS groups, indicating that the differentially expressed lncRNAs aptly reflected the differences between the BS and YS groups. It can also be concluded that the consistency of the differentially expressed lncRNAs in the six sequencing libraries was high, further reflecting the high accuracy and reliability of the sequencing data in this study.

Meanwhile, we identified 56 differentially expressed lncRNAs (|log2 (fold change)| ≥ 1, *p* < 0.05) between the BS and YS skin tissues. Among them (BS versus YS), 31 lncRNAs were upregulated, and 25 lncRNAs were downregulated (Figure 3f). We predicted the *cis* and *trans* targets of the differentially expressed lncRNAs to further examine how the lncRNAs function with the target genes to regulate the differences in skin color. In this study, the bioinformatic analysis revealed 3,417 cis-mRNA targets and 279 random trans-mRNA targets.

### 3.4. Examination of the Functions of the Differentially Expressed lncRNAs

It has been shown that lncRNAs can regulate the expression of neighboring coding genes and distal genes through *cis* [27] and *trans* [19] actions, respectively. To evaluate the potential *cis* regulatory functions of the lncRNA, we calculated Pearson correlation coefficients (PCCs) between the expression levels of lncRNAs and those of the neighboring protein-coding genes (lncRNA: cis-mRNA pairs). The positive regulation of peptidyl-tyrosine phosphorylation, protein tyrosine phosphatase activator activity, triose-phosphate isomerase activity, and dorsal/ventral pattern formation was revealed in the GO items (Figure 4a). Cytokine-cytokine receptor interactions, Herpes simplex infection, biosynthesis of amino acids, steroid hormone biosynthesis, oxidative phosphorylation, the MAPK signaling pathway, and arachidonic acid metabolism were the most significantly enriched pathways for the *cis* targets of the lncRNAs (Figure 4b). These results suggest that lncRNAs can modulate skin pigmentation by acting in *cis* fashion.

By *trans* target gene prediction, we identified many GO entries that are associated with melanogenesis, including protein tyrosine phosphatase activity, positive regulation of tyrosine phosphorylation of Stat3 protein, and skin development, among others (Figure 3c). The KEGG enrichment analysis showed that the *trans* target genes were enriched in 87 pathways, including six pathways that are known to be related to pigmentation: the MAPK signaling pathway, the Jak-STAT signaling pathway, the Wnt signaling pathway, the cAMP signaling pathway, the TGF-β signaling pathway, and melanogenesis (Figure 4d). The results show that the lncRNAs can also regulate melanin-related coding genes through trans action. A network analysis of the mRNAs and lncRNAs provides potential mRNA targets for differential lncRNAs. We found that the differential mRNAs endothelin 3 (*EDN3*), slowmo homolog 2 (*SLMO2*), and mitochondrial respiratory chain complex (*ATP5E*) were located near *LMEP* as the target genes, suggesting that *LMEP* may regulate skin melanin biosynthesis through adjacent melanogenic protein-coding genes (Figure 4e,f).

### 3.5. qRT-PCR Verification

To confirm the reliability of our RNA-seq data, nine genes (*ATP5E*, *SLMO2*, *TYR*, *EDN3*, *TCONS_00054154*, *TCONS_00070905*, *TCONS_00072039*, *Gallus_gallus_newGene_1037*, and *Gallus_gallus_newGene_720*) were chosen for the qRT-PCR, with β-actin serving as an internal reference gene. We found that the quantitative results of the genes were consistent with the sequencing results, indicating the accuracy of the sequencing results (Figure 5).

### 3.6. RACE, Subcellular Localization, Genomic Structure, and Protein-Coding Capabilities of LMEP

We cloned the 5’ and 3’ termini of *LMEP* by RACE PCR and confirmed that the sequence length of LMEP was 3093 bp. (Figure 6a,b). In the present study, we examined the subcellular localization of *LMEP* and found that *LMEP* was mainly expressed in the cytoplasm of melanocytes (Figure 6c,d). The location and exon distribution of *LMEP* in the genome were analyzed using the UCSC online database. The results showed that *LMEP* was located on chromosome 20 with six exons. Exon one was 222 bp, exon two was 115 bp, exon three was 95 bp, exon four was 76 bp, exon five was 70 bp, and exon six was 2506 bp in length (Appendix A). The coding ability of *LMEP* was predicted by CPAT and CPC, and we obtained prediction scores of 0.00420 and 0.0903 for *LMEP*, indicating that *LMEP* is a noncoding RNA (Appendix A). We found that *LMEP* is mainly localized in the cytoplasm according to the lncLocator prediction (Appendix A). It has been reported that many lncRNAs possess conserved secondary structures that form binding sites with protein molecules, which affect the regulation of the related gene expression. An analysis of the *LMEP* secondary structure using AnnoLnc shows many inner loops, convex loops and multibranch structures in the *LMEP* secondary structure (Appendix A).

### 3.7. Effects of LMEP Overexpression and Interference on Intracellular Tyrosinase

We observed the transfection of pcDNA3.1-LMEP-EGFP and siLMEP into melanocytes, and we detected of the relative expression of *LMEP* using qRT-PCR. We found that the expression of the *LMEP* gene was significantly higher in the treated group that was transfected with the pcDNA3.1-LMEP-EGFP recombinant plasmid than it was in the un-transfected plasmid blank group and the control group that was transfected with the pcDNA3.1-EGFP plasmid (*p* < 0.05) (Figure 7a). The expression of the *LMEP* gene in cells was significantly decreased after their transfection with siLMEP (*p* < 0.05), indicating a significant interference effect (*p* < 0.05) (Figure 7b).

We detected the intracellular tyrosinase content after the overexpression of the *LMEP* gene in melanocytes by an enzyme-linked immunosorbent assay. We found that the tyrosinase content in the treated cells that were transfected with the recombinant plasmid pcDNA3.1-LMEP-EGFP was significantly higher than that in the blank group without the plasmid transfection and the control group that was transfected with the pcDNA3.1-EGFP plasmid (*p* < 0.05) (Figure 7c). After interfering with the *LMEP* gene, the tyrosinase content in the treated cells that was transfected with siLMEP was significantly decreased when it was compared to the blank group and the control group that was transfected with NC (*p* < 0.05) (Figure 7d).

## 4. Discussion

Modern scientific research has confirmed that melanin serves many physiological functions, such as anti-oxidation, anti-aging, and anti-mutagenesis ones. Therefore, it is imperative to study the melanogenesis in black-bone chickens. The melanin synthesis process in the skin is complex. Genome-wide association studies (GWASs) and single nucleotide polymorphisms (SNPs) have been used to examine the pigmentary phenotypes [31,32,33]. With the development of high-throughput sequencing, more biologically significant lncRNAs have been discovered [5,12,13,34], which have functions such as regulating the gene expression and influencing the cell proliferation and differentiation of them. However, the lncRNAs in the skin tissue of black-bone chickens have been studied to a lesser extent.

Studies have shown that lncRNAs can interact with DNA, RNA, and proteins through *cis* and *trans* modalities. lncRNAs act in various ways, and they are divided into five main categories based on their functional characteristics: (1) The recruitment of chromatin remodeling complexes to repress or activate upstream and downstream gene expression [35,36]. (2) Their action as a decoy molecule to capture transcription factors and RNA polymerase II to silence target genes transcriptionally [37,38]. (3) The recruitment of proteins affecting variable splicing, mRNA stability, and translation processes [39,40]. (4) The lncRNAs affect the transcriptional activity of the target genes by competing with miRNAs for target gene binding sites or by binding to miRNAs [41,42]. (5) The lncRNAs can participate in the regulation of gene transcription and translation by encoding short functional peptides [43,44].

To better illustrate the potential mechanisms of the lncRNAs, we analyzed the functions of lncRNAs in dorsal skin and clarified whether the lncRNAs could act on protein-coding genes through the target genes [5,13,45]. The KEGG pathway analysis of the potential lncRNA targets revealed that the cytokine-cytokine receptor interaction, the MAPK signaling pathway, the Jak-STAT signaling pathway, the Wnt signaling pathway, the cAMP signaling pathway, the TGF-β signaling pathway, and melanogenesis were the most significantly enriched pathways. The involvement of these pathways in pigmentation has been previously validated. We found that lncRNAs act through the target genes in the melanogenesis-related pathways, but further experimental validation is needed.

In the present study, we found that *LMEP* acts on *EDN3*, *SLMO2*, and *ATP5E*, which are located on chromosome 20. *EDN3*, *SLMO2*, and *ATP5E* were located near *LMEP*, and the expression of these four genes was significantly higher in BS than they were in YS. A positive correlation was observed between the *LMEP* and *EDN3*, *SLMO2*, and *ATP5E* expression patterns. *EDN3* is mainly produced by keratinocytes in the epidermis and acts on adjacent melanocytes through paracrine pathways. Its primary function is to promote the proliferation, differentiation, and migration of melanocytes [46]. *EDN3* has two receptors, endothelin receptor B (EDNRB) and endothelin receptor B subtype 2E (EDNRB2), which can effectively promote the mitosis of the melanocytes to significantly increase the melanocyte numbers [47]. In mice, *EDN3* has been found to affect the melanin deposition in the skin; when *EDN3* is overexpressed in mice, the skin melanin also increases [48]. Studies on the pigmentation in humans [49], birds [50], cats [51], and other species have also shown that *EDN3* affects the synthesis and transport of pigment granules. The mitochondrial ATP enzyme ε subunit (ATP synthase H+-transporting mitochondrial F1 complex epsilon, *ATP5E*) is the smallest subunit of the catalytic center of the ATP synthase; *ATP5E* can bind with enzymes and inhibit the hydrolytic activity of ATP synthase through its functional domains, and it plays an essential role in the energy metabolism process [52,53]. Studies have shown that mutations in *ATP5E* cause mitochondrial energy metabolism disorders in higher eukaryotes [54]. In Silkie chickens, *ATP5E* and slowmo homolog 2 (*SLMO2*) are expressed at higher levels in the whole embryos of the FM phenotype breeds than they are in other breeds [46]. These results show that *LMEP* may play a regulatory role in skin melanin deposition through *EDN3*, *SLMO2*, and *ATP5E*.

In this work, we found a core lncRNA, *LMEP*, by integrated transcriptome sequencing. To verify the effect of *LMEP* on melanin deposition, we further performed a functional study of *LMEP* at the cellular level. The full-length *LMEP* sequence was obtained by a RACE assay, and a bioinformatics analysis was performed. The CPC and CPAT websites predicted that it has no coding ability and that it is a noncoding RNA. By isolating the nucleus and cytoplasm of the melanocytes and conducting a combined FISH detection, we found that *LMEP* was mainly localized in the cytoplasm. The synthesis of melanin is catalyzed by various intracellular enzymes. Tyrosine (TYR) is an aromatic amino acid precursor that catalyzes the synthesis of melanin from tyrosine [55]. TYR is a critical enzyme in the melanin synthesis pathway [9,56,57], and the expression and activity of TYR determine the rate and yield of the melanin synthesis. High levels of TYR promote melanin formation. To verify the effect of *LMEP* on melanin deposition, we transfected the *LMEP* overexpression vector and interfering fragment and measured the tyrosinase content by ELISA. The results showed an increase and decrease in the intracellular tyrosinase content after overexpression and interference, respectively, further suggesting that *LMEP* may be involved in regulating specific processes of pigmentation, which in turn promotes melanin deposition.

## 5. Conclusions

In summary, we elucidated differentially expressed lncRNAs in the skin of chickens of the same breed with different skin colors. The functions and biological processes that are associated with skin pigmentation-related lncRNAs were determined based on a bioinformatics analysis of the lncRNA target genes. The biological function identification of the core lncRNA *LMEP* further proved that *LMEP* might promote melanin deposition. These results and conclusions not only expand our understanding of lncRNA biology, but they may also provide valuable resources for future studies on skin pigmentation regulation by lncRNAs in chickens.

## Figures and Tables

**Figure 1 genes-13-02143-f001:**
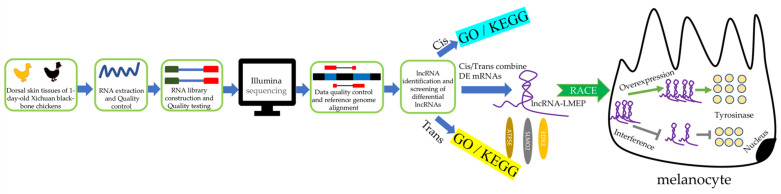
Flow chart for screening and functional validation of lncRNA-*LMEP*. In this article, we screened the core lncRNA-LMEP by transcriptome sequencing and validated its biological function. We found that lncRNA-LMEP may promote tyrosinase content through the *ATP5E*, *SLMO2* and *EDN3* genes to promote melanin deposition in chicken skin.

**Figure 2 genes-13-02143-f002:**
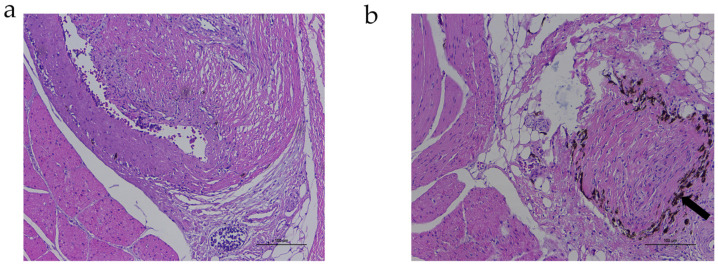
Histomorphological examination of YS and BS from Xichuan black-bone chickens. Differences in skin melanin deposition and histomorphology between YS (**a**) and BS (**b**). The arrow indicates melanin.

**Figure 3 genes-13-02143-f003:**
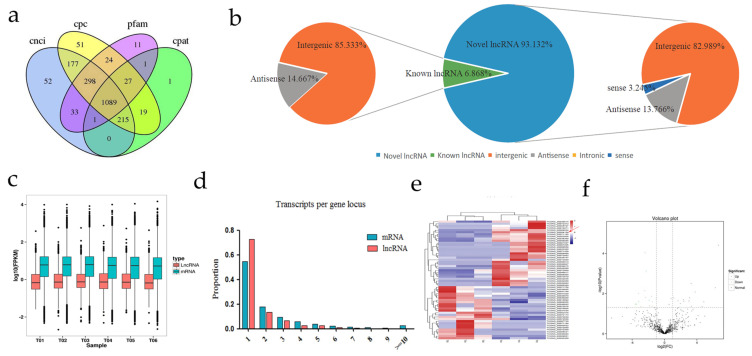
Identification of lncRNAs. (**a**) Screening candidate lncRNAs of the chicken skin transcriptome. Venn diagram of four lncRNA identification tools (CNCI, CPAT, CPC, and Pfam). (**b**) Classification of lncRNAs. (**c**) The horizontal and vertical axes represent the molecular type and log10 (FPKM), respectively. The boxes represent the quantile numbers. (**d**) The proportion of isoforms of lncRNAs and mRNAs at a different locus. (**e**) Hierarchical clustering of differentially expressed lncRNAs. (**f**) Volcano plot of the differentially expressed lncRNAs between YS and BS are shown.

**Figure 4 genes-13-02143-f004:**
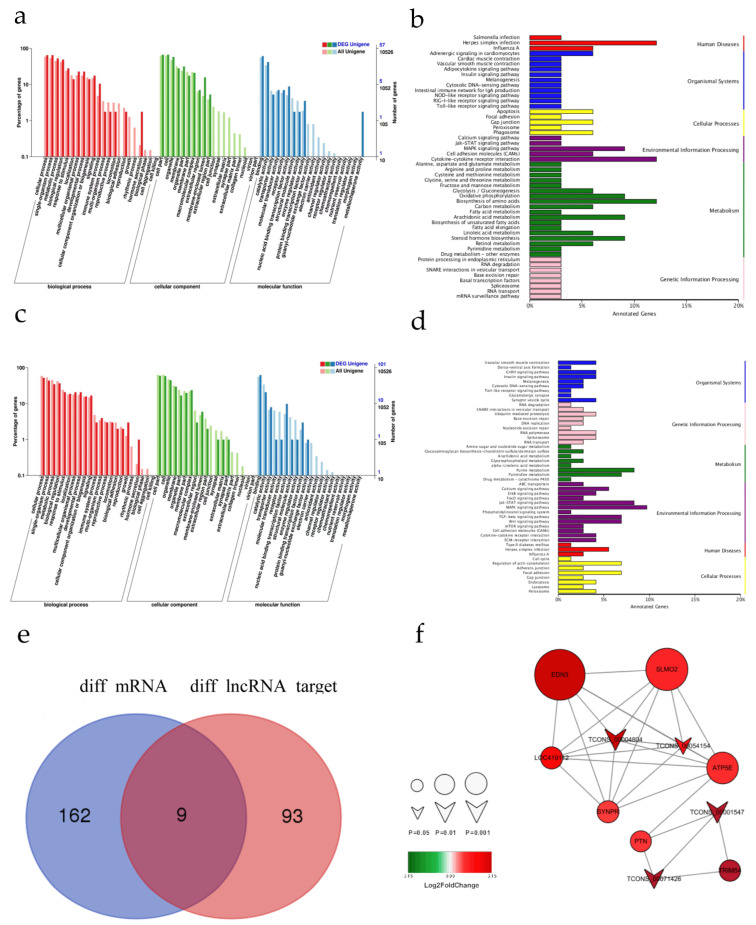
Functional enrichment of BS and YS tissue-derived lncRNA targets and comprehensive analysis of lncRNA-mRNA co-expression. (**a**) Comparison of the GO functional annotations of the *cis* target genes. (**b**) KEGG pathway annotations of *cis* target genes. (**c**) Comparison of the GO functional annotations of the *trans* target genes. (**d**) KEGG pathway annotations of the *trans* target genes. (**e**) Venn diagram of differentially expressed lncRNA target genes and differential mRNAs. (**f**) Differentially expressed lncRNA-mRNA network. Red and green indicate up- and downregulated genes, respectively. The shape size indicates logP; the smaller the *p* value is, then the larger the shape is.

**Figure 5 genes-13-02143-f005:**
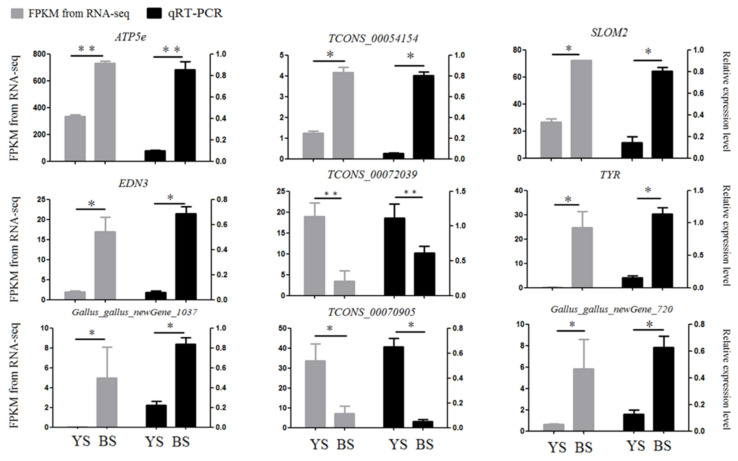
Quantitative validation of transcriptome sequencing results. * represents *p* < 0.05, ** represents *p* < 0.01.

**Figure 6 genes-13-02143-f006:**
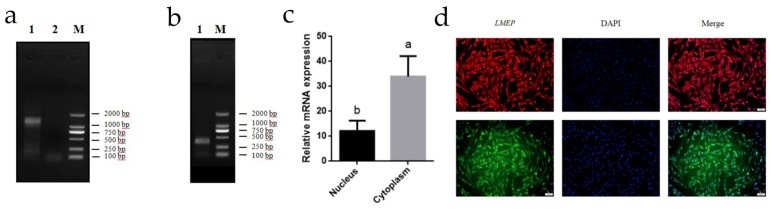
RACE and Subcellular Localization Identification of *LMEP*. (**a**) Results of *LMEP* full-length sequence amplification by RACE PCR. One and two represent 3′ RACE, while M represents DL2000 Markers. (**b**) One represents the 5′ RACE, while M represents DL2000 Marker. (**c**) Identification of the subcellular localization of *LMEP* by nuclear and cytoplasmic analysis. (**d**) FISH experiments to identify the subcellular localization of *LMEP*. The different letters represent significant differences (*p* < 0.05), and the same letters show no significant difference (*p* > 0.5), which is the same as it is below.

**Figure 7 genes-13-02143-f007:**
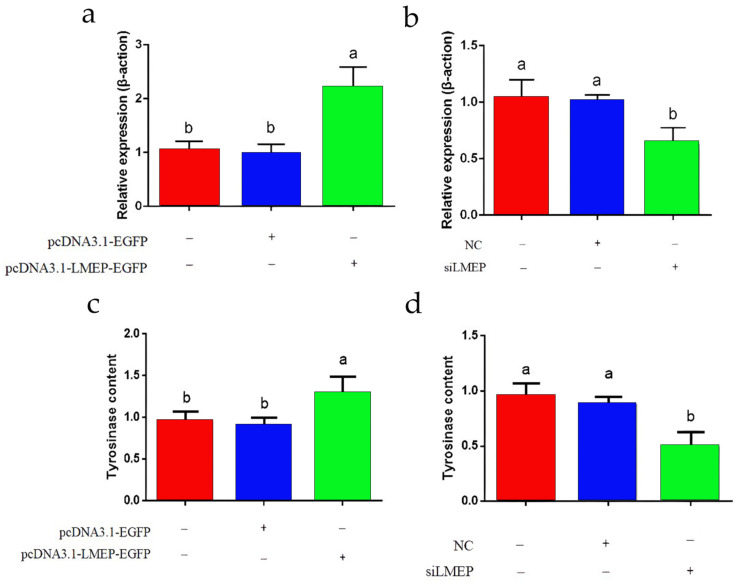
Effects of *LMEP* overexpression and interference on intracellular tyrosinase. (**a**) Detection of *LMEP* gene overexpression. (**b**) Detection of *LMEP* gene interference. (**c**) Effects of *LMEP* overexpression *LMEP* on tyrosinase. (**d**) Effects of interfering with *LMEP* on tyrosinase.

**Table 1 genes-13-02143-t001:** siRNA sequences of *LMEP*.

Name	Sense Sequence	Antisense Sequence
siNC siLMEP	UUCUCCGAACGUGUCACGUTT GGUGCUGUCACCCAUUGUUTT	ACGUGACACGUUCGGAGAATT AACAAUGGGUGACAGCACCTT

## Data Availability

The data were submitted to the Genome Expression Omnibus (Accession Numbers PRJNA418694) in NCBI.

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
