# Peer review of "Revealing the Regulatory Mechanism of lncRNA-LMEP on Melanin Deposition Based on High-Throughput Sequencing in Xichuan Chicken Skin"

_genes, 2022, doi:10.3390/genes13112143_

Round 1

Reviewer 1 Report

A good work, but need further revised as shown in the attachment.

Author Response

Dear Editor and Reviewers,

The authors thank the reviewers for the helpful comments and suggestions regarding our manuscript. We have made all necessary revisions and carefully addressed all comments as follows. In addition, we have carefully improved the English writing and checked the typos throughout the manuscript.

Q1: Lines 5-6: Please carefully check the superscript punctuation mark and special symbols of each author. There are many errors.

A1: Thank you for your comment. We surely agree with you. We have carefully checked the superscript punctuation marks and special symbols of each author and revised them accordingly.

Q2: Lines 21-22: The total length of LMEP should be stated clearly in the abstract.

A2: Thank you for your comment. We surely agree with you. We have stated the total length of LMEP in the abstract, such as” We found that the full length of the TCONS_00054154 sequence was 3,093bp by RACE PCR and named it LMEP.”

Q3: Lines 30-40: No reference(s) for the first paragraph? Please check.

A3: Thank you for your suggestion. We have double-checked the first paragraph and added the references.

Q4: Lines 44-48: The role of the content of tyrosinase in melanocyte in the formation of skin color shoul be mentioned in the second paragraph of the Introduction section. Because it is also one of the objects in your research.

A4: Thank you for your suggestion. We surely agree with you. We have added an introduction to tyrosine for skin tone formation in the second paragraph of the introduction, such as” Research on tyrosinase, also known as monophenol monooxygenase, shows that mutations in the tyrosinase gene can lead to abnormal skin pigmentation and result in skin diseases [8]. As a key enzyme in melanin synthesis, tyrosinase plays a crucial role in the complete process of production, and the activity and quantity of tyrosinase directly affect melanin production [9].”

Q5: Lines 62, 68, etc.: Check whether the space or full stop is before or after the position of references throughout the full manuscript.

A5: Thank you for your suggestion. We have checked to ensure that the space or full stop is before or after the position of references throughout the full manuscript; for example, we have revised "[]" to " []".

Q6: Lines 97-98: Please provide more detailed description of the specific location of the dorsal skin samples.

A6: Thank you for your comment. We surely agree with you. We obtained tissues with an area of 1.5 × 1.5 cm between the two dorsal wings. We described this as “tissue with an area of 1.5 × 1.5 cm between the two dorsal wings.”

Q7: Line 155: “Triton x-10” should be “Triton X-10”.

A7: Thank you for your suggestion. We have revised “Triton X-10” to “Triton X-100”.

Q8: Lines 158-159: Provide more details for “Washed at 42℃ and avoided light, DAPI stained, mounted, and fluorescence detection”. Such as regent, dosage, reaction conditions, etc.

A8: Thank you for your comment. We surely agree with you. We have added detailed steps for the reaction time, temperature and dosage, such as “The cells were washed 3 times at 42 °C for 5 min each to reduce the background signal, incubated with DAPI staining solution for 10 min at room temperature, and washed with PBS (phosphate-buffered saline) 3 times for 5 min each, followed by fluorescence detection. All procedures were performed under light-proof conditions.”

Q9: Lines 170-171: Miss the Table information for siRNAs

A9: Thank you for your suggestion. We have added siRNA-specific primer sequence information.

Q10: Lines 174-175: Provide detailed experimental steps for isolation and purification of primary melanocytes.

A10: Thank you for your comment. We surely agree with you. We have provided detailed experimental steps for the isolation and purification of primary melanocytes, such as” Chicken melanocytes were isolated from 20 embryonic Xichuan black-bone chickens. First, the eggs were wiped with 75% alcohol, the embryos were removed from the side of the air chamber, and the peritoneum was obtained by excision. The peritoneum was placed in prewarmed PBS with double antibodies at 37 °C and washed to remove grease and other impurities (repeated three times). The cleaned peritoneum was cut into pieces with ophthalmic scissors and washed three times with PBS, then placed in a 50 mL centrifuge tube with Medium 254 (Gibco, USA) before addition of an appropriate amount of Proteinase II enzyme and 0.25% trypsin digestion solution (1:1), digestion at 37 °C for 60 min (the suspended tissue floats and does not sink easily, the lump becomes flocculent when digestion is completed), and addition of complete medium to terminate the digestion. After repeated blowing, the samples were passed through a 200+400 mesh stainless steel sieve, and the filtrate was collected in a centrifuge tube. The samples were centrifuged at 1,000 r/min for 10 min, and the supernatant was discarded. Medium 254 was added, mixed with gentle blowing, and centrifuged at 1,000 r/min for 10 min, and the supernatant was removed (repeated 3 times). Fresh medium containing HMGS (Gibco, USA) (HMGS:Medium 254=1:100) was added, mixed well, and inoculated into cell bottles. The plates were incubated in 5% CO2 at 37 °C.

When melanocytes grew to 70%-80% confluence, the medium was discarded, the cells were washed with PBS 3 times, an appropriate amount of digestive solution (0.25% trypsinase-0.02% EDTA) was added, and the cells were digested at 37 °C for 7-10 min (when the cells were digested to approximately a half-rounded shape, complete medium was added to terminate digestion). After centrifugation at 1,000 r/min for 10 min, the supernatant was discarded, and fresh medium containing HMGS was added and cultured in an incubator with 5% CO2 and 37 °C. The solution was changed every 2 days.”

Q11: Line 185: “0.5 μL of the forward and ...” should be “0.5 μL each of the forward and ...”.

A11: Thank you for your suggestion. We have revised “0.5 μL of the forward and...” to “0.5 μL each of the forward and...”

Q12: Line 188: Mention the significance test for the results.

A12: Thank you for your suggestion. We have described the significance test for the results as follows: “We analyzed the quantitative data using the calculation of 2–ΔΔCt. Statistical analysis of the expression data for the different colors of chicken skin was performed with SPSS 26.0. One-way analysis of variance was used, and the results are expressed as the mean ± standard deviation (SD). Different lowercase letters or * indicate significant differences (P<0.05), and ** indicates a highly significant difference (P< 0.01).”

Q13: Lines 219-224: Figure 2 legend, “of lncRNAs (c)...” should be “of lncRNAs. (c) ...”. (e), Mention the samples for BS and YS in the figure legend. The color in Figure e indicates the expression level of different lncRNAs, but the expression pattern from different individuals in the same group is not consistent. Please explain this phenomenon? And use the arrow to indicate the location of LMEP.

A13 (1): Thank you for your suggestion. We have revised “of lncRNAs (c)...” to “of lncRNAs. (c)...”.

A13 (2): Thank you for your comment. We surely agree with you. Gene expression exhibits individual and tissue differences. Although there were differences in gene expression among individuals in our experiment, compared with the differences in gene expression between groups, the trend of gene expression among individuals in the group was consistent. Relate references literature: 1. Xu, E., Zhang, L., Yang, H., Shen, L., Feng, Y., Ren, M., & Xiao, Y. (2019). Transcriptome profiling of the liver among the prenatal and postnatal stages in chickens. Poultry science, 98(12), 7030–7040. https://doi.org/10.3382/ps/pez434. 2.Wu, P., Zhou, K., Zhang, J., Ling, X., Zhang, X., Li, P., Zhang, L., Wei, Q., Zhang, T., Xie, K., & Zhang, G. (2022). Transcriptome Integration Analysis at Different Embryonic Ages Reveals Key lncRNAs and mRNAs for Chicken Skeletal Muscle. Frontiers in veterinary science, 9, 908255. https://doi.org/10.3389/fvets.2022.908255. 3. Xu, Z., Che, T., Li, F., Tian, K., Zhu, Q., Mishra, S. K., Dai, Y., Li, M., & Li, D. (2018). The temporal expression patterns of brain transcriptome during chicken development and ageing. BMC genomics, 19(1), 917. https://doi.org/10.1186/s12864-018-5301-x. 4.Tang, R., Wang, J., Zhou, M., Lan, Y., Jiang, L., Price, M., Yue, B., Li, D., & Fan, Z. (2020). Comprehensive analysis of lncRNA and mRNA expression changes in Tibetan chicken lung tissue between three developmental stages. Animal genetics, 51(5), 731–740. https://doi.org/10.1111/age.12990. 5. Zhang, B., Yan, Z., Wang, P., Yang, Q., Huang, X., Shi, H., Tang, Y., Ji, Y., Zhang, J., & Gun, S. (2021). Identification and Characterization of lncRNA and mRNA in Testes of Landrace and Hezuo Boars. Animals : an open access journal from MDPI, 11(8), 2263. https://doi.org/10.3390/ani11082263. 6. Bao, G., Li, S., Zhao, F., Wang, J., Liu, X., Hu, J., Shi, B., Wen, Y., Zhao, L., & Luo, Y. (2022). Comprehensive Transcriptome Analysis Reveals the Role of lncRNA in Fatty Acid Metabolism in the Longissimus Thoracis Muscle of Tibetan Sheep at Different Ages. Frontiers in nutrition, 9, 847077. https://doi.org/10.3389/fnut.2022.847077 and other articles can illustrate this problem

A13 (3): Thank you for your suggestion. We have used the arrow to indicate the location of LMEP.

Q14: Lines 313-329: Merge Parts 3.7 and 3.8.

A14: Thank you for your suggestion. We have merged Parts 3.7 and 3.8.

Q15: Line 322: It’s confused that how to detect the content of intracellular tyrosinase. Please mention the method in “2. Materials and Methods” section.

A15: Thank you for your comment. We surely agree with you. We have supplemented the description of tyrosinase content in the Materials and Methods section, such as” Tyrosinase content was measured by an ELISA kit from Nanjing Jiancheng Bioengineering Institute (NanJing China). The experiment included a blank well, standard well and sample well, and each group included three replicates. Only chromogenic agent and stop solution were added to the blank well. Fifty microliters of standard and sample were added to standard and sample wells, respectively, followed by 50 µL of HRP, and incubated at 37 °C for 60 minutes. After discarding the solution, 200 µL of washing solution was added, washed for 30 s and patted dry, which was repeated 5 times. After adding the chromodeveloper, the treatment was kept in the dark at 37 °C for 10 min, and 50 µL of the stop solution was added to terminate the treatment. The OD value was detected at a wavelength of 450 nm.”

Q16: All the figures are blurry and can't be seen clearly. It is recommended to provide figures with high resolution that meet the requirements, and provide each figure with high resolution separately for later editing.

A16: Thank you for your suggestion. We have provided each figure with high resolution.

Q17: The Figure S3 is too blurry to see anything.

A17: Thank you for your suggestion. We have provided a clearer image.

Q18: Table S1: Please delete the column of “Primer size”, and add a column of “Tm” and “Product length” for primers, respectively.

A18: Thank you for your comment. We surely agree with you. We have deleted “Primer size” and added “Tm” and “Product length”.

Q19: Table S2: Did “NC” means “siNC”?

A19: Thank you for your suggestion. We have revised “NC” to “siNC.”

Q20: The tables lack notes.

A20: Thank you for your suggestion. We have added the notes.

Q21: Combing data from Table S1, Table S2 and Table S3 into one table, and add a column of “Application” to the right for showing the use of primer or base sequence.

A21: Thank you for your comment. We surely agree with you. We have merged Table S1, Table S2 and Table S3 into one table and added the “Application” column.

We appreciate for Editors/Reviewers’ warm work earnestly, and hope that the correction will meet with approval. Once again, thank you very much for your comments and suggestions.

Thank you very much.

If you have any further questions, please don’t hesitate to contact us.

Looking forward your response!

Yours sincerely,

Donghua Li, on the behalf of all the co-authors

College of Animal Science and Technology, Henan Agricultural University Zhengzhou 450046, China

Tel: +86-155-1697-6655

Reviewer 2 Report

well written research article, just check statistical analysis only

Author Response

Dear Editor and Reviewers,

The authors thank the reviewers for the helpful comments and suggestions regarding our manuscript. We have made all necessary revisions and carefully addressed all comments as follows. In addition, we have carefully improved the English writing and checked the typos throughout the manuscript.

Q1: average body weight and SD age of bird?

A1: Thank you for your comment. We surely agree with you. We used One-day old Xichuan black bone chicken with similar weight.

Q2: which experimental design was followed?

A2: Thank you for your suggestion. We have perfected the design of the experiment, such as” The results were expressed as mean ± SD (standard deviation) and analyzed by one-way ANOVA using SPSS (version 26.0, USA), and differences were considered significant at P value <0.05.”

Q3: Authors should add standard errors as it is better estimate, second add mathematical model for better understanding of analysis.

A3: Thank you for your comment. We surely agree with you. We learned that standard deviation is also widely used in data analysis. 1. Zhang, G., Wu, P., Zhou, K., He, M., Zhang, X., Qiu, C., Li, T., Zhang, T., Xie, K., Dai, G., & Wang, J. (2021). Study on the transcriptome for breast muscle of chickens and the function of key gene RAC2 on fibroblasts proliferation. BMC genomics, 22(1), 157. https://doi.org/10.1186/s12864-021-07453-0 2. Kui, H., Ran, B., Yang, M., Shi, X., Luo, Y., Wang, Y., Wang, T., Li, D., Shuai, S., & Li, M. (2022). Gene expression profiles of specific chicken skeletal muscles. Scientific data, 9(1), 552. https://doi.org/10.1038/s41597-022-01668-w 3. Li, S., Liu, R., Wei, G., Guo, G., Yu, H., Zhang, Y., Ishfaq, M., Fazilani, S. A., & Zhang, X. (2021). Curcumin protects against Aflatoxin B1-induced liver injury in broilers via the modulation of long non-coding RNA expression. Ecotoxicology and environmental safety, 208, 111725. https://doi.org/10.1016/j.ecoenv.2020.111725.

We appreciate for Editors/Reviewers’ warm work earnestly, and hope that the correction will meet with approval. Once again, thank you very much for your comments and suggestions.

Thank you very much.

If you have any further questions, please don’t hesitate to contact us.

Looking forward your response!

Yours sincerely,

Donghua Li, on the behalf of all the co-authors

College of Animal Science and Technology, Henan Agricultural University Zhengzhou 450046, China

Tel: +86-155-1697-6655

Reviewer 3 Report

Dear Authors,

The current research article is well organized with a strong scientific background. But my major concern is problem identification. The research outcome for molecular detection of skin colour is definitely scientifically sound or unique.  

It has been mentioned that the black skin colour of Xichuan chicken is of economic and medicinal importance. However the article lacks sufficient relevant supportive documents or references, thus justification of the importance of the problem is incomplete.  NGS sequencing is a rather expensive technique. Does it(molecular detection) justify the economic loss due to other than the black skin colour of the chicken? Appropriate references with figures, need to be provided to assess the economic loss.

The English language is not appropriate in most places. I need thorough editing. 

Author Response

Dear Editor and Reviewers,

The authors thank the reviewers for the helpful comments and suggestions regarding our manuscript. We have made all necessary revisions and carefully addressed all comments as follows. In addition, we have carefully improved the English writing and checked the typos throughout the manuscript.

Q1: It has been mentioned that the black skin colour of Xichuan chicken is of economic and medicinal importance. However the article lacks sufficient relevant supportive documents or references, thus justification of the importance of the problem is incomplete.  NGS sequencing is a rather expensive technique. Does it(molecular detection) justify the economic loss due to other than the black skin colour of the chicken? Appropriate references with figures, need to be provided to assess the economic loss.

A1 (1): Thank you for your comment. We surely agree with you. In China, most people like black bone chicken soup, and consumers will choose to buy black bone chicken with darker skin. Darker skin promotes consumer consumption and has higher economic value. The darker the skin is, the darker the muscles. The study found that black bone chicken muscle contains more carnosine and polyunsaturated fatty acids, is rich in nutrients, can provide a natural food supplement for people in poor health, has antioxidant and fatigue relieving effects, and has been applied in the treatment of wasting, anemia and diabetes and other diseases. Related references literature: 1. Tian, Y., M. Xie, W. Wang. H. Wu, Z. Fu, and L. Lin. 2007. Determination of carnosine in Black-Bone Silky Fowl (Gallus gallus domesticus Brisson) and common chicken by HPLC. Eur. Food Res. Technol. 226: 311-314. 2. Yong-gang Tu, Ya-zhen Sun, Ying-gang Tian, Ming-yong Xie, Jie Chen, Physicochemical characterisation and antioxidant activity of melanin from the muscles of Taihe Black-bone silky fowl (Gallus gallus domesticus Brisson), Food Chemistry 114: 1345-1450. 3. Yinggang Tian, Sheng Zhu, Mingyong Xie, Weiya Wang, Hongjing Wu, Deming Gong,Composition of fatty acids in the muscle of black-bone silky chicken (Gallus gellus demesticus brissen) and its bioactivity in mice, Food Chemistry 126: 479-483.

A1 (2): Thank you for your comment. We surely agree with you. RNA-seq is a widely used and relatively inexpensive technology. We can understand the mechanism of black skin formation through RNA-seq and other studies and develop molecular markers, which can solve the problem at a very low cost.

Q2: The English language is not appropriate in most places. I need thorough editing.

A2: Thank you for your comment. We surely agree with you. We apologize for our puzzling writing. We have improved the English writing throughout the manuscript.

We appreciate for Editors/Reviewers’ warm work earnestly, and hope that the correction will meet with approval. Once again, thank you very much for your comments and suggestions.

Thank you very much.

If you have any further questions, please don’t hesitate to contact us.

Looking forward your response!

Yours sincerely,

Donghua Li, on the behalf of all the co-authors

College of Animal Science and Technology, Henan Agricultural University Zhengzhou 450046, China

Tel: +86-155-1697-6655

Round 2

Reviewer 3 Report

Thanks for addressing the first query with relevant references. However they need to be included  within the research article. If possible, please add any quantitative data related to economic loss.

Author Response

Dear Editor and Reviewers,

The authors thank the reviewers for the helpful comments and suggestions regarding our manuscript. We have made all necessary revisions and carefully addressed all comments as follows.

Q1: Thanks for addressing the first query with relevant references. However they need to be included within the research article. If possible, please add any quantitative data related to economic loss.

A1: Thank you for your comment. We surely agree with you. In China, the production of black-bone chicken was 602,700 tons in 2015 and 786,500 tons in 2020, with a year-on-year growth of 4.95%. Meanwhile, the sales volume of black-bone chickens were 589,200 tons in 2015 and 770,100 tons in 2020, with a year-on-year growth of 6.03%. This shows that the growth rate of sales of black-bone chicken is higher than the growth rate of production. When we tried to improve the growth rate, we found the problem of reducing the blackness. After reducing the blackness, we are unable to sell them at the original price. Therefore, the exact loss cannot be estimated specifically because of the overall production and market fluctuations involved, but it is certain that the price for the same weight of black-bone chicken is 1~1.5 RMB/kg higher than that of broiler. The relevant data comes from https://www.sohu.com/a/540706486_121331963 and https://merchant.quanqinlian.com. Finally, we are very sorry that we did not find any quantitative data related to economic losses in the relevant English literature or in the Chinese literature. We found relevant quantitative data in the Chinese website, but could not provide the English website.

We appreciate for Editors/Reviewers’ warm work earnestly, and hope that the correction will meet with approval. Once again, thank you very much for your comments and suggestions.

Thank you very much.

If you have any further questions, please don’t hesitate to contact us.

Looking forward your response!

Yours sincerely,

Donghua Li, on the behalf of all the co-authors

College of Animal Science and Technology, Henan Agricultural University Zhengzhou 450046, China

Tel: +86-155-1697-6655
